# CONGA: Copy number variation genotyping in ancient genomes and low-coverage sequencing data

**Arda Söylev**[1,2]*, **Sevim Seda Çokoglu**[3], **Dilek Koptekin**[4], **Can Alkan**[5], **Mehmet Somel**[3]*

**1** Department of Computer Engineering, Konya Food and Agriculture University, Konya, Turkey, **2** Institute for Medical Biometry and Bioinformatics, Medical Faculty, Heinrich Heine University, Düsseldorf, Germany, **3** Department of Biology, Middle East Technical University, Ankara, Turkey, **4** Department of Health Informatics, Graduate School of Informatics, Middle East Technical University, Ankara, Turkey, **5** Department of Computer Engineering, Bilkent University, Ankara, Turkey

* arda.soylev@hhu.de (AS); msomel@metu.edu.tr (MS)

## Abstract

To date, ancient genome analyses have been largely confined to the study of single nucleotide polymorphisms (SNPs). Copy number variants (CNVs) are a major contributor of disease and of evolutionary adaptation, but identifying CNVs in ancient shotgun-sequenced genomes is hampered by typical low genome coverage (<1×) and short fragments (<80 bps), precluding standard CNV detection software to be effectively applied to ancient genomes. Here we present CONGA, tailored for genotyping CNVs at low coverage. Simulations and down-sampling experiments suggest that CONGA can genotype deletions >1 kbps with F-scores >0.75 at $\geq$1×, and distinguish between heterozygous and homozygous states. We used CONGA to genotype 10,002 outgroup-ascertained deletions across a heterogenous set of 71 ancient human genomes spanning the last 50,000 years, produced using variable experimental protocols. A fraction of these (21/71) display divergent deletion profiles unrelated to their population origin, but attributable to technical factors such as coverage and read length. The majority of the sample (50/71), despite originating from nine different laboratories and having coverages ranging from 0.44×-26× (median 4×) and average read lengths 52-121 bps (median 69), exhibit coherent deletion frequencies. Across these 50 genomes, inter-individual genetic diversity measured using SNPs and CONGA-genotyped deletions are highly correlated. CONGA-genotyped deletions also display purifying selection signatures, as expected. CONGA thus paves the way for systematic CNV analyses in ancient genomes, despite the technical challenges posed by low and variable genome coverage.

## Author summary

In parallel with developments in genomic technologies over the last decades, ancient genomics opened a new era in understanding the evolutionary history of populations and species. However, the field still needs novel computational methods for accurate and

**Data Availability Statement:** All relevant data are within the manuscript and its Supporting information files. CONGA is implemented in C

programming language and its source code is available under BSD 3-clause license at https://github.com/asylvz/CONGA, as well as Supplemental Code. Simulated datasets and predictions of each tool can be accessed through Zenodo (10.5281/zenodo.5555990). Mappability data was downloaded from http://hgdownload.cse.ucsc.edu/goldenpath/hg19/encodeDCC/wgEncodeMapability/.

**Funding:** This work was supported by the European Research Council Consolidator grant "NEOGENE" (Project No.:772390 to MS) https://erc.europa.eu/. The funders had no role in study design, data collection and analysis, decision to publish, or preparation of the manuscript.

**Competing interests:** The authors have declared that no competing interests exist.

effective use of ancient genome data, which is mostly low-coverage and more challenging to analyse than modern-day genomes. Single nucleotide polymorphisms (SNPs), to date, have yet been the main source of information analysed in ancient genome studies. This is despite copy number variants (CNVs) harboring at least as much information as SNPs, especially with respect to natural selection. Here we developed CONGA, an algorithm for genotyping deletions and duplications in low-coverage genomes. We assessed its accuracy using simulations (with ancient-like data), and also studied its performance among 71 real ancient human genomes from different laboratories. We found that the common practice of authors filtering their ancient genome data before publishing prevents the reliable identification of duplications. Meanwhile, large (>1,000 base-pair) deletions can be detected even at quite low coverage (e.g. 0.5×). Deletions called in ancient genomes reflect population history and also show signs of negative selection.

This is a *PLOS Computational Biology* Methods paper.

## Introduction

Ancient genomics, the analysis of genetic material extracted from archaeological and paleonto-logical remains, has become a major source of information for the study of population history and evolution over the last decade [1–4]. While the number of published ancient genomes is exponentially growing, their analyses have yet been nearly exclusively limited to those of single-nucleotide polymorphisms (SNPs), whereas structural variations (SVs) in ancient genomes remain mostly ignored. Copy number variations (CNVs) are a common type of SVs and include deletions and duplications ranging from 50 bps to several megabasepairs. Although their number, by count, is much fewer than SNPs, the fraction of the genome affected by CNVs is well past that accounted by SNPs [5]. Likewise, CNVs are a major contributor to phenotypic variation: they are frequently discovered as the basis of diverse biological adaptations [6–14] as well as genetic diseases (reviewed in [15–18]). This renders the study of CNVs in ancient genomes two-fold attractive. First, as CNVs frequently serve as genetic material for adaptation, their study in ancient genomes can allow detailed temporal investigation of adaptive processes. Examples include evolutionary changes in salivary amylase copy numbers in humans and in dogs, thought to represent responses to a shift to starch-rich diets [19, 20]. Second, large deletions can be a major source of deleterious mutation load, and studying deletion frequencies in ancient genome samples from extinct species or severely bottlenecked populations can inform about the genetic health of lineages. For instance, a study on the last surviving mammoth population on Wrangel Island reported an excess of deletions in this sample, which may have compromised the population's fitness [21].

Despite this appeal, the impact of CNVs on evolutionary history and ancient phenotypes remains largely unexplored [2]. The reason lies in the significant technical challenges in CNV detection posed by ancient genomes. State-of-the-art methods for CNV discovery from shotgun genome sequencing data require at least moderate depth of coverage [22–25] and read-pair information [26–31], or long reads [32, 33]. However, due to the degraded and elusive nature of ancient DNA, ancient genome data is frequently produced at low coverage (<1×) and the molecules retrieved are typically short, between 50–80 bps. Excess variability in genome coverage

caused by taphonomic processes is another potential issue. Although CNVs have been studied in a few relatively high coverage ancient genomes using CNV discovery tools [20, 21, 34–36], these methods are inapplicable to most ancient genome data sets, and so far, no specific algorithm for CNV identification in ancient genomes has been developed and tested.

With the aim to fill this gap, here we present CONGA (**Co**py **N**umber Variation **G**enotyping in **A**ncient Genomes and Low-coverage Sequencing Data), a CNV genotyping algorithm tailored for ancient and other low coverage genomes, which estimates copy number beyond presence/absence of events. We use simulations and down-sampling experiments to assess CONGA's performance. Beyond simulations, we explore whether deletions can be reliably genotyped in heterogeneous datasets composed of ancient genomes from different laboratories, where not only low coverage, but also coverage variability caused by differences in taphonomy and experimental protocols may pose challenges. We evaluate this by studying expected patterns of genetic drift and negative selection on CONGA-genotyped deletions.

## Results

### Motivation and overview of the algorithm

We developed CONGA to genotype given candidate CNVs in mapped read (BAM) files (Methods). The choice of CNV genotyping over CNV discovery has obvious reasons: (a) CNV discovery using low coverage ancient genomes is impractical; (b) for many species studied using ancient genomics, CNV reference sets based on high quality genomes are already available (Supplemental Note 1 in S1 Text); (c) variants in ancient genomes will largely overlap with present-day variants in most cases; (d) genotyping has much shorter running times and lower memory usage than discovery. Indeed, although algorithms for *de novo* SNP discovery exist [37, 38], most ancient genome studies to date have chosen genotyping known variants because of low coverage and DNA damage [39]. We reasoned that it may be likewise possible to genotype CNVs in ancient genomes with high accuracy and in short running times using depth of coverage and split-read information, despite low and variable coverage.

Briefly, CONGA first calculates the number of reads mapped to each given interval in the reference genome, which we call "observed read-depth". It then calculates the "expected diploid read-depth", i.e., the GC-content normalized read-depth given the genome average. Using these values, CONGA calculates the likelihood for each genotype by modeling the read-depth distribution as Poisson, similar to some common CNV callers [40–42]. The genotypes can be homozygous CNV, heterozygous CNV, or no CNV. Using these likelihoods CONGA then calculates a statistic we term the C-score, defined as the likelihood of a CNV being true (in heterozygous or homozygous state) over it being false (no CNV). For genotyping duplications, CONGA also uses an additional split-read step in order to utilize paired-end information. Briefly, it splits reads and remaps the split within the genome, treating the two segments as paired-end reads [31, 43]. Either type of signature, read-depth or paired-end, can be sufficient to call a duplication (Methods). The overall workflow is presented in Fig 1. We note that an alternative CNV genotyping tool, GenomeSTRiP [44, 45], also uses similar information but is mainly designed for genotyping multiple genomes simultaneously, and evaluates the read depth data using Gaussian mixture models instead of Poisson.

### Accuracy evaluation using simulated genomes and comparison with published algorithms

To evaluate the performance of CONGA we first simulated ancient-like genomes with CNVs. We employed VarSim [46] to insert deletions and duplications into the human reference

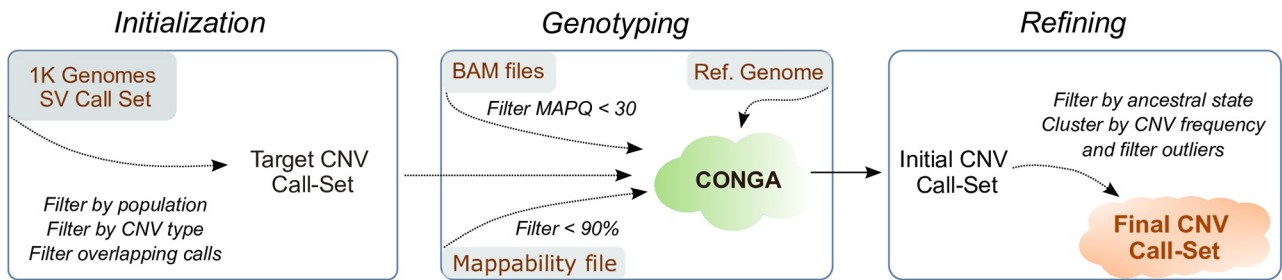

**Fig 1. Overall workflow of CONGA.** The first step involves initialization, where we create the input (reference) CNV file using deletions and duplications identified in high quality genome sets. We apply CONGA-genotyping in the second step and create the initial CNV call set. We then perform filtering and refining steps, and thus generate the final CNV call set.

genome (GRCh37). We used three different size intervals for CNVs: small (100 bps–1000 bps), medium (1,000 bps–10,000 bps) and large (10,000 bps–100,000 bps). We thus simulated three genomes, each with roughly 1,500 deletions and 1,500 duplications of a specific size range (see S1 Fig for the exact numbers and length distributions of CNVs inserted in each genome). We next used these genomes as input to the ancient read simulator Gargammel [47], which generates paired-end short Illumina reads with varying fragment sizes (median 66 bps) and post-mortem damage. The data was generated at various depths: 0.05×, 0.1×, 0.5×, 1× and 5× (Methods). We then used CONGA to genotype CNVs across the simulated ancient genomes, using a candidate CNV call set. In order to assess specificity and sensitivity, we prepared the candidate CNV call set to include both true and false events. The false (background) CNV list was prepared using published deletion and duplication calls from modern-day human long-read sequencing datasets [48–51], as well as from African populations (AFR) from Phase 3 of the 1000 Genomes Project [52]. We mixed false and true CNVs with a ratio of approximately 10:1 (∼15,000 false events vs. ∼1,500 true events) and used this mixed list as the candidate CNV call set to CONGA (Methods). To assess the performance of CONGA in identifying CNVs, we further compared it with GenomeSTRiP [44, 45] and three of the widely used CNV discovery tools: CNVnator [22], FREEC [23] and mrCaNaVaR [25, 53, 54]. Fig 2 summarizes the comparison performance of CONGA, GenomeSTRiP, FREEC and CNVnator using precision-recall curves and F-score plots for deletions and duplications using various depths of coverage for medium and large CNVs. In Table A in S1 Table, we further present the number of true and false predictions by each tool, as well as their true positive rates (TPR), false discovery rates (FDR) and the F-scores (F1) for identifying deletions and duplications of small, medium and large size (as defined above). We also provide the results of mrCaNaVaR predictions for large variations, as this algorithm was specifically designed to target >10 kbps duplications only.

Our results revealed that both genotypers, CONGA and GenomeSTRiP, achieve higher performance compared to the three CNV discovery tools (Fig 2 and S1 Table). Although the superior performance of callers versus discovery tools is expected, the result is still non-trivial given the fact that our candidate CNV call set included 10 times more false CNVs than true CNVs, marking the specificity achieved by both CONGA and GenomeSTRiP.

CONGA and GenomeSTRiP had comparable performances, although CONGA had lower FDR and slightly lower recall (TPR) than the latter, leading to overall higher F-scores (Fig 2 and Table A in S1 Table). We note that GenomeSTRiP was run on each genome independently here, and its performance could be higher if multiple genomes were genotyped together [44].

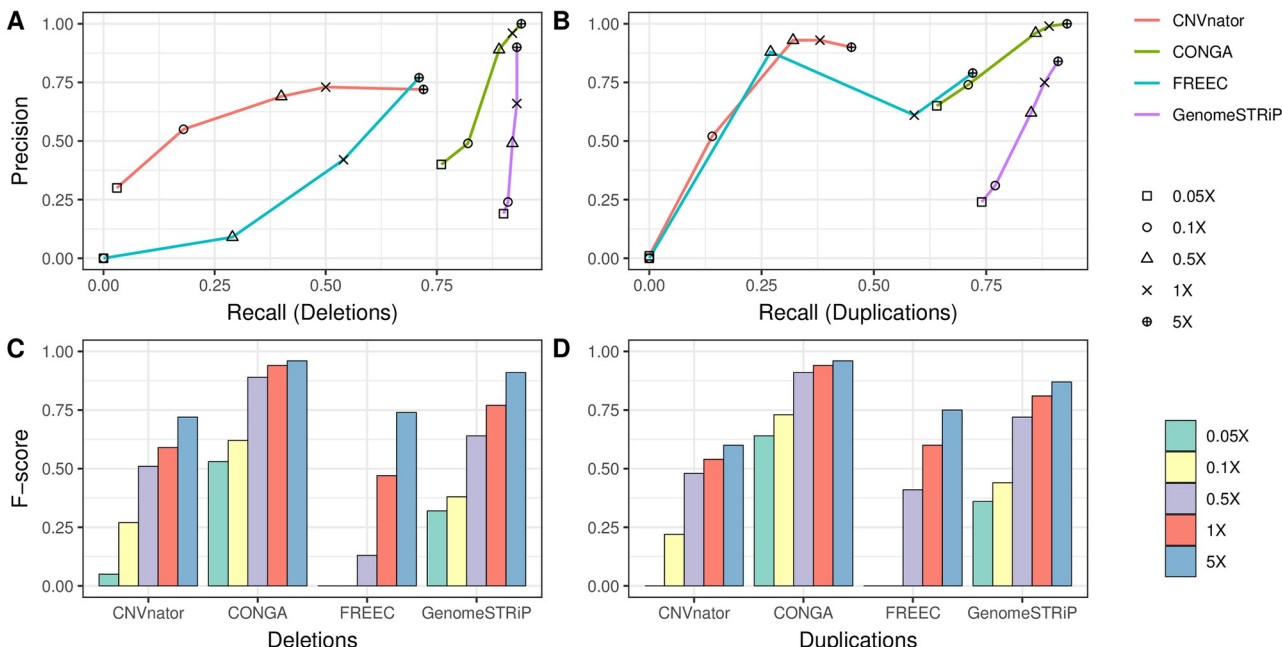

**Fig 2. CNV prediction performances of CONGA, GenomeSTRiP, FREEC and CNVnator on simulated genomes.** In (A) and (B), we show recall-precision curves based on depths of coverage values for deletions and duplications, respectively. In (C) and (D), we show the F-scores, calculated as (2 × *Precision* × *Recall*)/(*Precision* + *Recall*)). The figures represent the average statistics calculated for medium (1 kbps–10 kbps) and large (10 kbps–100 kbps) CNVs. See Table A in S1 Table for detailed information including small (100 bps–1 kbps) CNVs, as well as mrCaNaVaR predictions for large variations. Commands used to run each tool are provided in S1 Text. The results here were generated using the cutoff C-Score <0.5 for CONGA, while no read-pair or mappability filters were applied.

However, joint genotyping may also create biases in heterogeneous datasets comprising genomes produced using different protocols (see Discussion).

We observed that all tools converge in performance as the coverage approaches depths of 5×, especially with large CNVs. For small CNVs (<1 kbps), all tools under-performed, although CONGA predictions still had higher recall and precision than the other tools (see S2 Fig for precision-recall curves).

The simulation results thus suggest that CONGA can efficiently and accurately genotype deletions and duplications of length >1 kbps in ancient genomes at ≥0.5× coverage, with higher overall accuracy compared to available discovery and genotyping tools.

**Diploid genotype inference.** Beyond the identification of deletion and duplication events, classifying individual genotypes as heterozygous or homozygous CNVs could provide valuable information for population genetic analyses of CNVs. However, predicting CNV copy numbers on low coverage genomes can be a significant challenge [55]. We thus assessed the performance of CONGA in determining the diploid genotype of a CNV event based on the likelihood model described above. We employed the same simulation data, and focused on medium and large size CNVs given the weak performance of CONGA on small CNVs. We note that CONGA only evaluates the possibility of homozygous duplications (ignores copy numbers ≥3). Fig 3 shows CONGA's performance in determining diploid genotype states for >1 kbps deletions and duplications for each coverage tested. We found that F-scores were ≥0.70 at coverages ≥0.5×. Encouragingly, CONGA had comparable power in identifying both heterozygous and homozygous events (Table C in S1 Table).

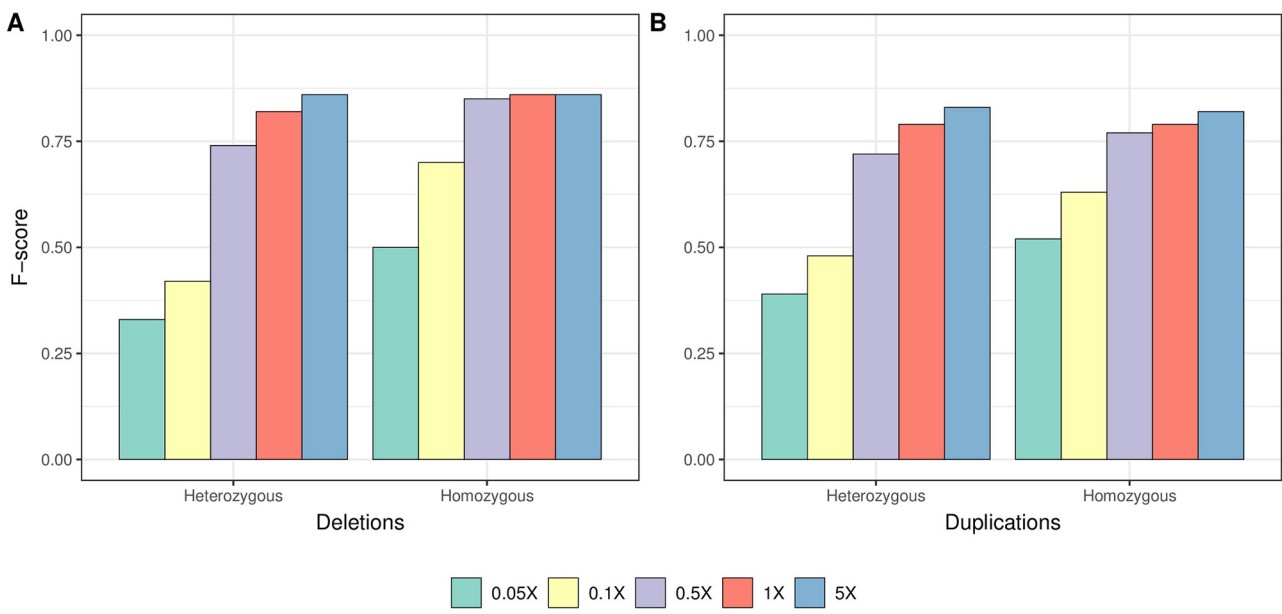

**Fig 3. Performance of CONGA in correctly inferring copy-numbers.** The figure shows the genotype (heterozygous vs homozygous) prediction performance CONGA for (A) deletions and (B) duplications using merged sets of medium and large CNVs, at various coverage values.

## Down-sampling experiments with real ancient genomes

We next studied the performance of CONGA in identifying CNVs at various depths of coverage using real ancient genome data. As no ground truth CNV call-set is available, we used the following approach: (i) we chose three published ancient genomes of relatively high coverage ($\geq$9×), (ii) we genotyped CNVs using the full genome data with CONGA and using a modern-day human CNV call set as input, (iii) we down-sampled the ancient genome data to lower coverages, (iv) we assessed CONGA's performance in genotyping the same CNVs at low coverage (Methods).

Specifically, we selected a ($\sim$ 23.3×) ancient Eurasian genome (Yamnaya) [56], a 13.1× ancient genome from Greenland (Saqqaq) [57], and a 9.6× ancient genome from Ethiopia (Mota) [58]. The Yamnaya genome was only available as a BAM file, while the latter two were available as FASTQ files, which we processed into BAM files (Methods). We used a list of modern-day human CNVs as candidate CNV set ($n = 17, 392$ deletions and $n = 14, 888$ duplications) (Methods) as input to CONGA. We thus genotyped between 688–1,581 deletions and 638–4,097 duplications across these three genomes using the full data. We then down-sampled all three BAM files to various depths, and repeated the genotyping for each genome. We estimated CONGA's TPR and FDR on down-sampled genomes by treating the CNVs genotyped using the full data as ground truth (Methods).

CONGA displayed satisfactory performance in identifying deletions in all three genomes even at coverages around 0.5×, with TPR of >70% and FDR of <45% (Fig 4 and Table D in S1 Table). For duplications, however, CONGA showed poor performance: at around 1× coverage, duplication TPR was >40% in the Saqqaq and Mota genomes, and only 22% in the Yamnaya genome. A detailed analysis of these results suggested that pre-publication quality filtering of the Yamnaya genome BAM files may have obliterated read-depth-based duplication signals in the data (Supplemental Note 2 in S1 Text).

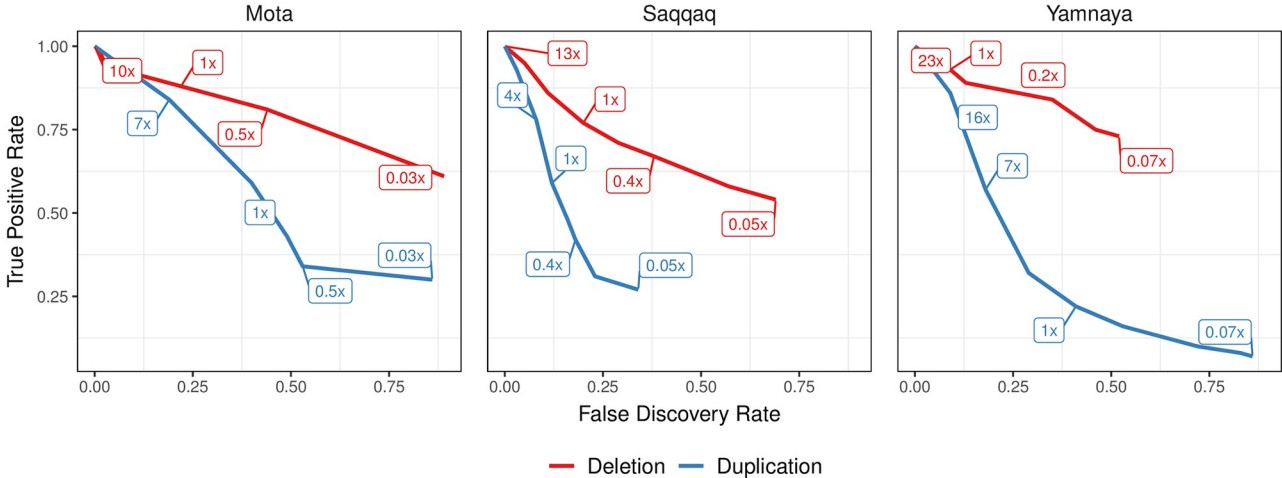

**Fig 4. TPR vs FDR curves for deletion and duplication predictions of CONGA.** Here, we use Mota, Saqqaq and Yamnaya genomes down-sampled to various depths from their original coverages of 9.6×, 13.1× and 23.3×, respectively. The numbers inside boxes show the down-sampled coverage values. We calculated TPR and FDR for down-sampled genomes assuming that our CONGA-based predictions with the original genomes (full data) reflect the ground truth. These predictions, in turn, were made using modern-day CNVs as candidate CNV list. The purpose of the experiment was to evaluate accuracy at lower coverage relative to the full data, as well as to compare performance across different real genomes (Methods).

Overall, both our simulations and down-sampling experiments with real genomes suggest that CONGA can efficiently genotype >1 kbps deletion events at depths of coverage of 0.5×, and possibly even at 0.1×. CONGA could thus be applied on a large fraction of ancient shotgun sequenced genomes available for deletion genotyping. In contrast, CONGA's low performance in duplication genotyping in the down-sampled Yamnaya BAM data implies that identifying duplications in published low coverage ancient genomes may not be feasible, as to date, such data are mainly submitted to public repositories in BAM format (see Discussion). We therefore limited downstream analyses on real ancient genomes to deletions >1 kbps.

## Analysis of 71 real ancient genomes reveal technical influences on deletion genotyping

Although CONGA's above performance in deletion genotyping was promising, heterogeneous sets of real ancient genomes may pose additional challenges, as they are obtained from DNA samples of variable taphonomic history and are produced via different experimental protocols. Hence, whether consistent biological signals may still be extracted from low coverage genome sets remains unclear. To explore this, we genotyped deletions with CONGA across a diverse sample of real ancient human genomes. We then studied their diversity with the expectation that deletions, like SNPs, should display genome-wide similarity patterns that reflect population origin, i.e., shared genetic drift, among individuals [59–61].

We thus collected BAM files for 71 ancient human genomes belonging to a time range between c.2,800–45,000 years Before Present (BP) (S2 Table) [56, 62–86]. These were chosen to bear diverse characteristics, including mean coverage (0.04×-26×, median = 3.45×), population origin (West Eurasia, East Eurasia, and North America), the laboratory of origin (10 different laboratories), the use of shotgun vs. whole-genome capture protocols, or the use of uracil–DNA–glycosylase (UDG) treatment [87]. For genotyping, we employed a candidate CNV dataset of 11,390 autosomal deletions (>1 kbps with mean 10,735 bps) identified among African populations (AFR) from Phase 3 of the 1000 Genomes Project [52] (Methods). Our

motivation for using an African sample here was to avoid ascertainment bias [88] in studying deletion frequencies, as all of the 71 ancient individuals were non-African, and thus African populations represent an outgroup to our sample set. We further filtered these for high mappability (mean per locus >0.9) and to be derived in the human lineage (using chimpanzee and bonobo genomes to represent the ancestral state), leaving us with 10,002 deletion events (Methods).

Genotyping the 10,002 loci across 71 BAM files, we found 8,780 (88%) genotyped in at least one genome (as deletion or reference). Of these, 5,467 (55%) loci were genotyped as a deletion in heterozygous or homozygous state. Across the 71 genomes, we detected a median number of 490 deletion events [396–2, 648] again in either heterozygous or homozygous state.

We then studied the frequencies of these deletion across the 71 ancient genomes using a battery of heatmaps, hierarchical clustering, multidimensional scaling plots (MDS) and principal components analysis (PCA) (S3 and S4 Figs). This revealed a minority of genomes exhibiting highly divergent frequencies, without obvious association with their population of origin. Given the close evolutionary relationship among Eurasian human populations, we reasoned that these divergent signals most likely originate from experimental artifacts, data processing artifacts, or variability of DNA preservation among samples. Accordingly, mean deletion frequencies across the 71 genomes could be at least partly explained by laboratory-of-origin (Kruskall-Wallis test, $p = 0.08$).

Studying the data further, we identified a subset of 21 divergent, or outlier genomes (S3 and S4 Figs). Removing these also removed the laboratory-of-origin effect (Kruskall-Wallis test, $p = 0.22$; Supplemental Note 3 in S1 Text). Moreover, we could recognize a number of attributes that could explain these divergent deletion profiles. First, the 21 divergent genomes had on average shorter read length compared to the remaining 50 genomes (median = 57 vs. 69; Wilcoxon rank sum test $p < 0.001$; S3(A) Fig). One of these was the Iceman, with unusually short (50 bps) reads. Second, the coverage of the 21 divergent genomes was lower compared to the rest (median = 3.31 vs. 3.98; Wilcoxon rank sum test, $p = 0.014$; S5(B) Fig). For instance, all three genomes with <0.1× coverage in our dataset (ne4, ko2, and DA379) were among the outliers. The number of non-genotyped loci was likewise higher in the divergent group (median = 1509 vs. 1886; Wilcoxon rank sum test, $p = 5.39 \times 10^{-5}$; S5(C) Fig). Meanwhile, UDG-treatment did not appear to be related to outlier behaviour (binomial test $p = 2.633 \times 10^{-9}$; S5(D) Fig). Finally, Bon002, the only sample produced using whole-genome capture, was among the most extreme outliers, implying that capture distorts coverage. We consequently removed these 21 genomes from further analyses, thus retaining only shotgun-sequenced genomes with coverage >0.4×.

## A comparison of deletion and SNP diversity across 50 ancient genomes

The above filtering steps resulted in a dataset of 8,780 derived deletions genotyped in at least one of the 50 ancient Eurasian genomes, with 396–748 deletions (median = 467.5) detected in heterozygous or homozygous state per genome, and 29% detected in at least one genome.

We used this dataset to test three hypotheses: (i) that CONGA-called deletion diversity patterns parallel SNP diversity patterns, reflecting shared demographic history (genetic drift and admixture) among genomes, (ii) that CONGA-called deletions should be evolving under some degree of negative selection (caused by gene expression alterations, exon loss, or frame-shifts), and (iii) that variation in deletion load among genomes may be correlated with variation in deleterious SNP load. We note that the first two patterns (hypotheses i and ii) have been previously described using large modern-day CNV datasets (see [59–61] for drift, and [5, 89, 90] for

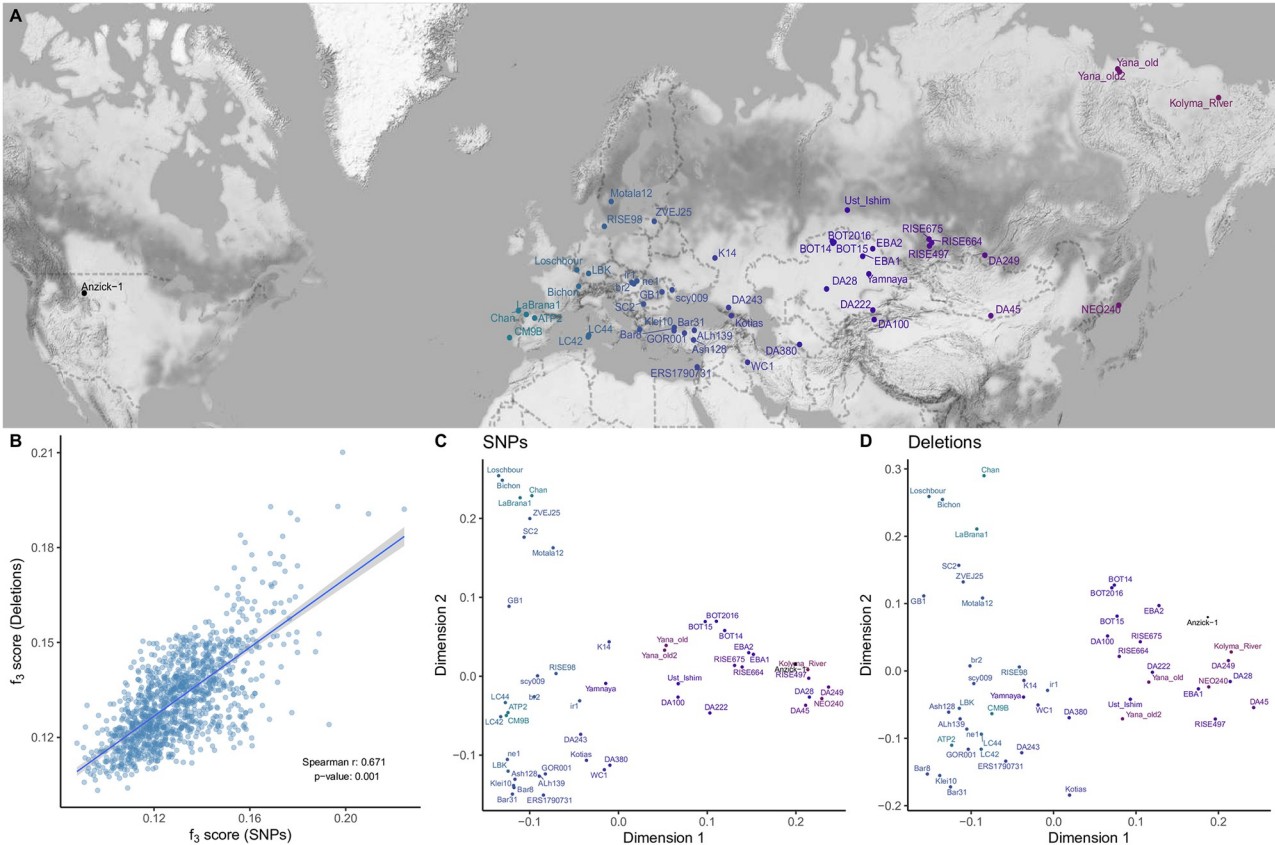

**Fig 5. Deletion vs. SNP diversity.** (A) Geographic locations of the 50 ancient individuals used in the analyses. (B) Comparison of genetic distances calculated between pairs of genomes using SNPs and deletions. We calculated the Spearman correlation coefficient between two matrices and then calculated the Mantel test p-value using the "mantel" function in R package "vegan" (v2.5–7). (C) and (D) represent multidimensional scaling plots that summarize outgroup-$f_3$ statistics calculated across all pairs among the 56 ancient individuals using SNPs and deletions, respectively. To create the map, we used Stamen map tiles (Map tiles by Stamen Design, under CC BY 3.0. Data by OpenStreetMap, under ODbL) with "ggmap" [92] R package. We added points and texts to plots by using "ggplot2" [93] and its extension "ggrepel" [94].

selection), and our goal here was mainly to perform a sanity check and assess the effectiveness of CONGA in producing reliable biological signals.

To test the first hypothesis, we compared pairwise genetic distances among the 50 individuals (Fig 5A) calculated using either SNPs or deletion genotypes. For this, we collected 38,945,054 autosomal SNPs ascertained in African individuals in the 1000 Genomes Dataset and genotyped our 50 ancient genomes at these loci (Methods). We then calculated pairwise outgroup-$f_3$ statistics, a measure of shared genetic drift between a pair of genomes relative to an outgroup population [91]. Using the Yoruba as outgroup, we calculated genetic distances for all pairs of ancient genomes as $(1—f_3)$, using either SNPs or deletions. We observed strong positive correlation between the two resulting distance matrices (Spearman $r = 0.671$, Mantel test $p = 0.001$) (Fig 5B). Summarizing SNP- and deletion-based distances using multidimensional scaling (MDS) also revealed highly similar patterns, with clear clustering among west and east Eurasian genomes observed with either type of variation (Fig 5C and 5D). This result was encouraging in showing that diversity patterns based on deletion genotyping across heterogeneous and low coverage ancient genomes can still reveal expected signals of shared demographic history.

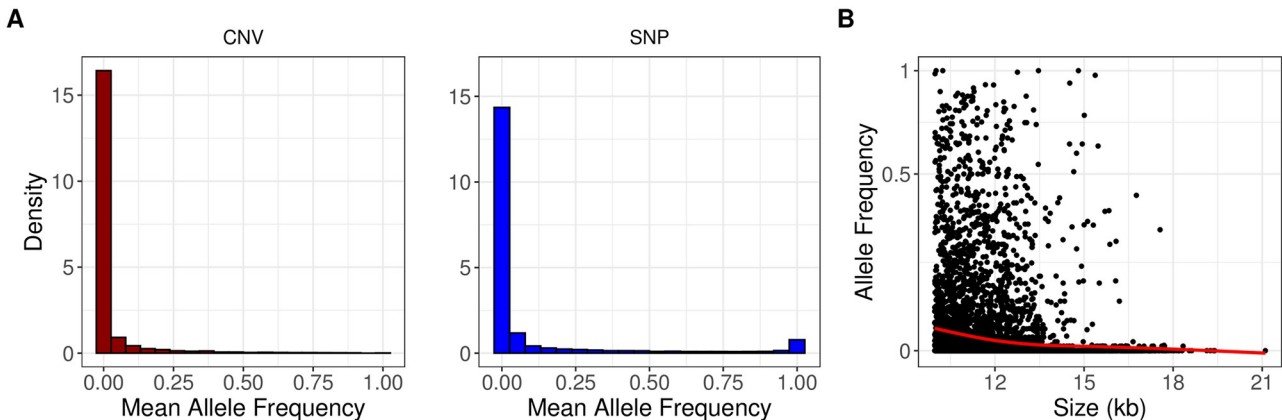

**Fig 6. The impact of negative selection on deletions.** (A) The site frequency spectra of derived deletion alleles (on the left, n = 3,472) and derived SNP alleles (on the right, n = 57,307). Both plots represent frequency distributions of loci genotyped in all 50 ancient genomes (no missing data). Zero indicates lack of derived alleles. The two distributions are significantly different from each other (Kolmogorov-Smirnov test $p < 10^{-15}$). (B) The size distribution (in kbps) of deletion events versus allele frequency. The red line shows the smoothing spline fit and indicates a negative correlation (Spearman correlation $r = -0.33$, $p < 10^{-16}$). Both axes were $log_2$-scaled.

## Negative selection on deletion variants

We next studied the impact of negative (purifying) selection on deletions by comparing the site frequency spectrum (SFS) of autosomal deletions with those of SNPs. We used the 8,780 human-derived deletions and 32,304,437 human-derived SNP alleles across the 50 ancient Eurasian genomes (Methods). To allow comparison with the pseudo-haploidized SNP genotype data, we randomly chose one allele per genome (i.e., deletion or no event) in the deletion dataset. Set side by side with the SNP SFS, we observed an excess of undetected derived deletions and singletons, and a lack of fixed derived variants among deletions, consistent with stronger negative selection on the latter (Fig 6A). The excess of undetected and singleton deletions does not appear to be related to low recall, as both high and low coverage genomes show the same trend (S6 Fig).

If deletions are under negative selection we may also expect longer deletions, or deletions containing evolutionary conserved genes, to be segregating at lower frequencies, similar to that observed with modern-day human genomes [5, 89, 90]. Indeed, we found that deletion allele frequencies were negatively correlated with deletion size across the 50 ancient genomes (Spearman correlation $r = -0.33$, $p < 10^{-16}$) (Fig 6B). To test the second idea, we determined deletions overlapping Ensembl human genes. Overall, 26% of the 8,780 derived deletions overlapped minimum one gene. We then collected mouse-human dN/dS ratios, an inverse measure of protein sequence conservation (Methods). We found that deletions with lower (below-mean) allele frequency showed a tendency to disrupt genes with lower dN/dS values compared to deletions with higher (above-mean) allele frequency (median = 0.086 vs. 0.097; Mann-Whitney U test, one-sided $p = 0.055$). These observations, along with the SFS comparison, follow the notion that deletions are evolving under negative selection.

We further asked whether inter-individual variation in the total deletion mutation burden measured here may be correlated with variation in the burden of functional deleterious SNPs based on their impact on protein sequence. Demographic bottlenecks can theoretically cause variable levels of mutation burden –as deletions and/or as SNPs– among populations, and these burden levels could be correlated especially if their phenotypic impacts are comparable (see Discussion). To test this we collected (a) total deletion length and (b) the number of genes

affected by deletions, for each of the 50 ancient Eurasian genomes (S7 Fig). We further collected SIFT scores (an estimate of how protein sequence would be affected by a SNP [95]) for n = 22,996 SNPs in our dataset, predicted to be "deleterious" or "tolerated", and used these to calculate a deleterious/tolerated ratio per genome (Methods; A in S8 Fig). We then compared the deleterious/tolerated ratio-based burden levels with deletion-based mutation burden levels (using either total deletion length or the number of deletion-affected genes), but found no significant correlation (Spearman $r = 0.09$ and $r = -0.05$, respectively, $p > 0.5$; B in S8 Fig). This could be explained by high noise and lack of statistical power, as well as differences in phenotypic impacts between deletions and SNPs (see Discussion). We also did not observe any correlation between historical age and deletion frequencies in this sample (S9 Fig).

## Time and memory consumption

Finally, we examined time and memory requirements of CONGA and other tools using the same computing resources (Intel Xeon CPU E5–2640 v2 @ 2.00 GHz: 2 CPUs * 8 cores each and 216 GB RAM; single threaded). We first tested CONGA's deletion genotyping performance on the BAM files of the 71 ancient genomes presented above. This finished in ∼12 hours in total, with as low as 2.2 GB of peak-memory consumption. This translates to ∼10 minutes per genome. In order to evaluate CONGA's performance with a higher coverage genome sample, we ran 30 genomes (randomly selected 10 CEU, 10 YRI, 10 TSI) from the 1000 Genomes Project Phase 3, which had mean 7.4× coverage [52]. The analysis took just slightly longer, ∼15 minutes average per genome, with similar memory usage.

We also compared the time and memory requirements of CONGA, GenomeSTRiP, FREEC and CNVnator in Table 1. In order to benchmark these tools, we used a 5× simulated genome (the same genome with medium sized CNVs used in the simulation experiments described above). CONGA has the lowest runtime and memory footprint among the other tools.

In Table E in S1 Table, we report the effects of parameter choices on CONGA's runtime and memory usage. We note that using split-reads for duplication genotyping (intended for higher coverage genomes) increases runtime and memory consumption significantly because here CONGA uses its own small-scale read mapper, which creates a bottleneck.

We further provide a comparison of CONGA's performance on genomes of various depths of coverage in Table E in S1 Table, calculated using the down-sampled 23× Yamnaya genome (with coverages between 23× and 0.07×).

**Table 1. Time and memory consumption.**

| Tools | Time (h:m) | Peak Memory Usage (GB) |
|---|---|---|
| CONGA | 0:09 | 1.2 |
| GenomeSTRiP | 1:22 | 2.2 |
| FREEC | 0:39 | 7.1 |
| CNVnator | 0:32 | 14.1 |

Time and memory consumption of each algorithm for a simulated genome of 5× depth of coverage with 1680 deletions and 1684 duplications. "Time" refers to wall clock time and "Peak memory usage" is the maximum resident set size. Note that GenomeSTRiP has two steps in its pipeline: preprocessing and genotyping. Here, time was calculated by summing the running times of each step, and memory by taking the maximum. For CONGA, we used default parameters used in the simulation experiments.

## Discussion

Modern human genome sequencing experiments today typically reach coverages >20× and increasingly use long read technology, and such experiments can employ diverse read signatures to reliably identify CNVs [96]. CONGA's approach, which mainly relies on the read-depth signature, is naive in comparison; however, using read-depth appears as the most practical solution given the short fragment size and the predominance of low coverage (around or <1×) among ancient genome datasets.

### CONGA's overall performance and utility

Despite these challenges, our experiments using simulated genomes and down-sampled real ancient FASTQ data showed that CONGA can relatively efficiently genotype deletions and duplications of size >1 kbps at 1× coverage, or even lower. CONGA outperformed two "modern DNA" CNV discovery algorithms, FREEC and CNVnator, which have been previously employed in ancient genome analyses [97, 98]. CONGA exceeded both tools in TPR and true negative rates, especially at coverages <1×. This may be unsurprising, as these tools were developed for discovering novel CNVs in relatively high coverage genome data. Meanwhile, compared to GenomeSTRiP, a CNV genotyper that also uses both different sources of information within a Bayesian framework [44, 45], CONGA performed better in achieving lower FDR rates at all coverages, while GenomeSTRiP had higher recall at coverages 0.5× or below. We also found that 96% of deletion and 89% of duplication predictions of CONGA overlap with those of GenomeSTRiP (calculated as the average match rate of medium and large sized CNVs) (Table A in S1 Table). The predictions by GenomeSTRiP match those of CONGA at lower rates (c.55%), mainly owing to higher FDR in GenomeSTRiP results. In time and memory use, CONGA surpassed all three tools.

In terms of deletion copy number estimates, CONGA again achieved acceptable accuracy (∼75% TPR and <30% FDR) in genomes of 0.5× coverage. At lower depths of coverage and also when genotyping deletions <1 kbps, recall and/or precision were weaker. CONGA's performance on duplications was also poor, as we discuss below.

Overall, the relatively high accuracy at ≥0.5× coverage suggests that CONGA could be used to genotype deletions across a considerable fraction of published shotgun sequenced ancient genomes. CONGA and GenomeSTRiP could also be used in parallel, as they appear to complement each other in recall and specificity. Further, GenomeSTRiP can be used in population samples for jointly genotyping low coverage genomes, which could potentially increase performance. We caution, however, that joint genotyping can create ascertainment biases if coverage and ancestry co-vary among jointly analysed genomes.

Beyond aDNA, CONGA is suitable for CNV analyses for any low depth whole-genome sequencing (WGS) experiment. Such studies are increasing in number due to the trade-off between budget limitations and the wealth of genome-wide information that can be used in population and conservation genetics (e.g. [99]).

### Constraints in duplication genotyping

In simulated genome experiments, CONGA's performance in genotyping duplications was similar to that in deletions. This performance was again mainly based on read-depth information, while paired-end information was also effective at 5× coverage (Table F in S1 Table and S10 Fig).

CONGA also showed slightly lower but comparable performance in duplication genotyping than in deletion genotyping when analysing the Mota and Saqqaq genomes, both available as FASTQ files. However, CONGA's performance was dramatically low on the 23× ancient BAM

file, Yamnaya. This can be explained as follows (see Supplemental Note 2 in S1 Text): (i) The available Yamnaya data was filtered in such a way that excess reads at duplicated loci, i.e. read-depth information, was lost. (ii) Consequently, nearly all (97%) duplications CONGA geno-typed in the original (23×) BAM file were called only using paired-end information. (iii) Because paired-end information is rapidly lost with decreasing coverage (as it requires reads overlapping breakpoints), and read-depth information was lacking, genotyping duplications in this BAM file became infeasible at <5× coverage.

The majority of shotgun ancient genomes in public databases are only available as BAM files. The majority of published files are also at <5× coverage. Hence, most published ancient shotgun genomes yet are not amenable to duplication genotyping with CONGA. This is highly unfortunate, as gene duplications are a major source of evolutionary adaptation that would be valuable to study also in ancient populations.

### Constraints in deletion genotyping

Applying CONGA to genotype deletions on a heterogeneous set of real ancient shotgun genomes revealed conspicuous technical influences on deletion genotyping, with a major fraction of the 71 analysed genomes displaying outlier behaviour in their deletion frequencies. We could notice technical particularities for the 21 genomes identified as outliers, such as lower coverage, shorter read lengths, or the application of whole-genome hybridization capture. Our results suggest that 0.4× coverage may be close to the lower threshold for deletion genotyping of >1 kbps events, slightly higher than the threshold in our simulation results. We also find that whole-genome hybridization capture and extra short reads (roughly ≤55 bp) compromise deletion genotyping, while UDG-treatment does not show a significant effect. That said, we lack clear explanations for outlier deletion frequency patterns for some of these 21 genomes. For instance, the genome SI-45 has coverage >3× and an average read length of 60 bps, but nevertheless displays unusual deletion patterns. We suspect that such unexpected patterns might reflect technical peculiarities in library preparation, sequencing or data filtering. Unique taphonomic processes influencing DNA preservation and variability in coverage may also be at play.

Such effects could be investigated by future studies compiling larger datasets with detailed experimental descriptions. Meanwhile, our results point to the necessity of rigorous quality control and outlier filtering when calling deletions in heterogeneous datasets, similar to practices traditionally adopted in transcriptome analyses. This is particularly essential when combining genomes produced using different experimental protocols and sequencing platforms.

### Community recommendations for improving CNV analyses in ancient genomes

The above observations mark the need for an update of practices in producing and publishing ancient genomes to allow reliable study of both deletions and duplications, beyond SNPs.

- Most published ancient genome data to date is SNP capture data, which is largely worthless for CNV analyses. Our results underscore the long-term value of shotgun sequencing data over SNP capture, as well as whole-genome capture.

- Publishing data as raw FASTQ files should be priority. The main motivation behind publishing BAM files instead of raw data is to avoid publishing environmental DNA reads, which constitute a large fraction of reads from shotgun sequenced aDNA experiments. Saving microbial (e.g. pathogenic) aDNA fractions for future investigation is another motivation. Nevertheless, our results demonstrate that raw FASTQ data is absolutely necessary for

duplication genotyping at low coverage and also helpful against biases in deletion genotyping. In the long term, publishing raw data will be for the whole community's benefit.

- Sharing all details on DNA extraction, library construction, as well as the alignment and pre-processing steps used in creating the exact version of datasets submitted to public databases is crucial for healthy reuse of the data.

### Purifying selection and mutation loads in past populations

Our analysis of >1 kbps deletions genotyped in 50 ancient genomes revealed how variation in deletion frequencies reflect (a) demographic history, as visible in strong correlation with SNP variation and spatial clustering, and (b) negative selection, as seen in a steeper SFS than of SNPs, lower frequencies of large deletions, and lower frequencies of deletions overlapping conserved genes. These results demonstrate that CONGA can identify reliable biological signals in technically heterogeneous and noisy datasets.

Beyond expected patterns, we also studied possible correlation between deletion loads and deleterious SNP loads per genome across the 50 ancient individuals. High deleterious mutation loads could arise by relaxation of negative selection due to strong bottlenecks, as suggested for Wrangel Island mammoths [21] or for dogs [100]. Conversely, bottlenecks can cause high inbreeding levels, and this may lead the purging of recessive deleterious variants, as recently described for a founder population of killer whales [101]. In our dataset we found no significant relationship between deletion-related loads and deleterious SNP loads. This could be due to lack of strong variability among Eurasian genomes in deleterious mutation burdens or due to low statistical power, as we only use deletions segregating in Africa. The result could also reflect differences in dominance effects or fitness effects between SNPs and deletions.

A full analysis of this question could be possible with the creation of a geographically comprehensive genomic time-series, including genomes of non-Eurasian populations with variable demographic histories. It would further require CNV discovery in carefully processed high-coverage ancient genomes and subsequent genotyping on low coverage data using CONGA. We hope that our study opens the way for such work, bringing deeper insight into the impacts of selection and drift in humans and other species.

### Methods

Among various approaches developed for CNV discovery using high throughput sequencing data, almost all use the fact that read-depth, i.e., the density of reads mapped to the reference genome, will be lower in deleted regions and higher in duplicated regions relative to the genome average [96, 102]. The distance between paired-end reads, their orientation, and split-read information (start and end of reads mapping to different locations) are further sources of information used in determining CNVs. Although available CNV discovery algorithms generally perform well in modern-day human genome sequencing data with high coverage, this is not necessarily the case for ancient genomes, as well as other low coverage sequencing experiments (S11 and S12 Figs). The first reason is that the majority of shotgun ancient genomes are produced at low coverage (typically <1×), which constrains the use of read-depth information. Second, ancient DNA fragments are short and of variable size (typically between 50–100 bps) [3]. Thus, paired-end information is absent, and available split-read information is also limited. Variability in ancient DNA preservation and genome coverage [103] is yet another noise source that is expected to limit efficient CNV discovery. CONGA overcomes these limitations using genotyping instead of *de novo* discovery. It estimates whether a candidate CNV, the location of which is provided as input, is present in a genome in BAM format. It also estimates the

diploid genotype, i.e., the heterozygous or homozygous state. CONGA makes use of read-depth information for deletions, and both read-depth and split-read information for duplications. We provide the details of our split-read approach in Supplemental Note 4 in S1 Text.

## Likelihood-based read-depth calculation for deletion and duplication genotyping

The input to the algorithm is (1) a list of candidate CNV locations and CNV type, i.e., deletion or duplication, and (2) a data set of reads aligned to the linear reference genome (e.g. using BWA [104]), which should be in BAM format.

In order to calculate the likelihood of a CNV at a given locus based on read-depth information, CONGA uses an approach akin to [31]. Let $(S_i)$ be the $i^{th}$ CNV in our CNV input list, defined by the breakpoint interval $(B_l, B_r)$ and the type of CNV: a deletion or duplication. At this locus, CONGA calculates the likelihood of the three possible genotype states, $k$, given the read alignment data and CNV type. These states are: no event ($k = 0$), a heterozygous state ($k = 1$), or a homozygous state ($k = 2$). The likelihood, in turn, is calculated by comparing the observed ($O_i$) read-depth versus the expected ($E_{ik}$) read-depth within $(B_l, B_r)$, given the three different genotypes. We detail the steps below.

1. We count the total number of mapped reads within that locus (falling fully within the interval $(B_l, B_r)$). This is the observed read-depth, $(O_{RD})$.

2. We calculate expected read-depth under a "no event" scenario, i.e., representing the diploid state. Here we account for the GC bias in high-throughput sequencing data [105], by using LOESS smoothing to normalize read-depth for GC content. Specifically, for each chromosome, we calculate the read-depth values per GC percentile for sliding windows of size 1,000 bps (step size = 1 bp). We then calculate the average read-depth per GC percentile. Then, using the chromosome-wide average GC value for the interval $(B_l, B_r)$, we calculate the expected diploid read-depth, $E_{i_{k=0}}$.

3. We model the read-depth distribution as Poisson, using the expected read-depth values for $k = 0$, $k = 1$, $k = 2$. We calculate the probability $P(RD_{S_i}|state = k)$ as:

$$P(RD_{S_i}|state = k) = \frac{E_{ik}^{O_i} \times e^{-E_{ik}}}{O_i!},$$

where $E_{i_k}$ is the expected read-depth given $state = k$, and $O_i$ is the observed read-depth at that specific locus. A typical autosomal human locus is diploid (has copy number 2); therefore when there is no CNV event ($k = 0$), the expected value of $O_i$ should be $E_{i_{k=0}}$.

If a genome is homozygous for a deletion, we expect no reads mapping to the region, thus $O_i \sim E_{i_{k=2}} = 0$. For heterozygous deletions, the expected number of mapped reads in that interval will be half of the expected diploid read-depth: $O_i \sim E_{i_{k=1}} = E_{i_{k=0}}/2$. For homozygous duplications, we expect $O_i \sim E_{i_{k=2}} = E_{i_{k=0}} \times 2$. For heterozygous duplications, we expect $O_i \sim E_{i_{k=1}} = E_{i_{k=0}} \times 1.5$.

4. We calculate a likelihood-based score, which we term the C-score, to estimate how likely locus $S_i$ carries a non-reference variant in a genome, in one or two copies. For this we use the calculated likelihoods for the three states. We define the C-score as the maximum of the likelihoods of $(S_i)$ being present in heterozygous state ($k = 1$) or in homozygous state ($k = 2$) in that genome, over the likelihood of no event ($k = 0$). We use the log function to avoid

numerical errors.

$$C - score(S_i) = \frac{max(log(P(RD_{S_i}|k = 1)), log(P(RD_{S_i}|k = 2)))}{log(P(RD_{S_i}|k = 0))},$$

The C-score is distributed between 0 and $+\infty$, with lower scores indicating higher likelihood of a true CNV event.

Results from our simulations and down-sampling experiments suggest that the relatively simple Poisson distribution can be effectively used to model copy number states, especially in the face of potentially non-independent errors due to ambiguous mapping of short and damaged reads or GC content heterogeneity. We note that alternative models have also been used for analysing CNVs in short read sequencing data, such as the negative binomial distribution [106] or Gaussian mixed models [44]. We also note CONGA's our approach could be expanded in the future by including the evaluation of duplication events involving >2 copies, as in multicopy genes [107].

## Mappability filtering

The probability of unique alignment of a read of certain size varies across the genome, mainly due to repetitive sequences. Various algorithms estimate this probability, termed mappability, across the genome for k-mers of specific length [108–111]. This is calculated by extracting k-mers of given length through the genome, remapping them to the reference genome, and measuring mappability as the proportion of unique mappings [110]. Because low mappability regions can be confounded with real deletions, we use mappability information to filter out CNV loci that could represent false positives.

CONGA accepts any mappability file in BED format, where values are distributed between 0 and 1. These can then be used to filter out CNVs for minimum mappability.

In our experiments, we used the 100-mer mappability data from the ENCODE Project [112] (see Software Availability). Using this data, for each CNV event ($S_i$), we calculated the average mappability value within its breakpoints. We used a minimum average mappability threshold of 0.9 for the CNV events we analyzed.

Our deletion frequency analysis results suggest strict mappability filtering (e.g. applying >0.9) should be used especially when analyzing data sets of heterogeneous origin. This is because published BAM files frequently differ in mapping quality filters applied before publishing (and these filters are usually not indicated). Such filtered BAM files will produce artificial deletion signals at low mappability regions, while unfiltered BAM files will not.

## Overlapping CNV events

In performing CNV estimations we filtered pairs of overlapping events from the input set (i.e. the call set). Specifically, if two CNVs overlaped >50% of their size, we excluded the smaller event.

There are two main reasons for this filtering step. First, CNV breakpoint resolution of CNV callers can frequently be imprecise, such that overlapping CNVs might actually be representing the same event called with slightly different breakpoints (e.g. with 100 bp distance) on different genomes or with different tools. Second, allowing overlapping CNVs in the input set (i.e. the call set) creates the risk of calling multiple events although only one event is actually present. For instance, if a small deletion resides within a larger deletion, a genome carrying the larger deletion in heterozygous state would be automatically genotyped heterozygous for the smaller deletion. This could introduce various biases in CNV estimates and down-stream analyses. We

thus preferred to be conservative and removed overlapping events, although we note that our calls could still be affected by real overlapping events not in our input list, and we would be also missing some events from our analyses. This is a point that could be addressed in future work.

## Simulation and down-sampling experiments

**Simulating ancient genomes with implanted deletions and duplications.**  Our goal here was to study the performance of CONGA on different sized deletions or duplications using simulated genomes containing implanted CNVs and to determine thresholds for reliably calling these variants. We first employed VarSim [46] to simulate and insert deletions and duplications into the human reference genome GRCh37. We repeated this three times, for small (100 bps–1000 bps), medium (1000 bps–10,000 bps), and large (10,000 bps–100,000 bps) CNVs. As a result we generated three CNV-implanted genomes, with around 1500 deletions and 1500 duplications each (between 1385 and 1810). The CNVs were produced so that they were non-overlapping, and their length distribution and exact counts are provided in S1 Fig.

To evaluate specificity and sensitivity, we also included a background (false) CNV set in the experiment, which would not be implanted but would be queried as part of the candidate list. This background set was prepared using recently published deletion and duplication calls from human genome sequencing experiments [48–51] and also sequencing data from African populations (AFR) from Phase 3 of the 1000 Genomes Project [52]. We compiled a list of 17,392 deletions and 14,888 duplications that were non-overlapping and of size $> \sim 1000$ bps using BEDTools mergeBed [113]. When evaluating genomes with small CNVs (100 bps–1,000 bps), we additionally included small CNVs from [49]. Specifically we added 4,623 deletions and 3,750 duplications of size 100 bps–1,000 bps to the above background list.

In order to assess CONGA's performance, we added the true CNVs generated using VarSim to this background set (and removed overlapping CNVs from the candidate genotype set), such that only $\sim 10\%$ of the input candidate CNV list were true events. Finally, we determined how many of these true events could be correctly called by CONGA and other methods.

**Simulating ancient genome read data.**  We used the above-described simulated genomes as input to Gargammel [47], which generates ancient-like Illumina reads, i.e., short reads of variable size bearing postmortem damage (i.e., C-to-T transitions at read ends) and including adapters. Gargammel can generate aDNA fragments following a size distribution given as input, and we used a subset of [77], which is default for this software. We used Gargammel to produce reads at various depths of coverage: 0.05×, 0.1×, 0.5×, 1× and 5×. We then removed adapters and merged overlapping reads [114] to generate single-end Illumina reads. These reads had sizes ranging between 34 bps and 139 bps, with average 69 bps and median 66 bps (these statistics were calculated using 1× coverage data, but other data also had similar distributions). We mapped the Gargammel-output reads back to the human reference genome (hg19, or GRCh37) using BWA-aln [104] with parameters "-l 16500 -n 0.01 -o 2". Note that BWA-aln has been shown to be more accurate for short ancient reads than BWA-mem [115].

**Evaluation of CONGA, GenomeSTRiP, CNVnator and FREEC with simulated ancient genome data.**  We ran CNVnator [22], FREEC [23] and GenomeSTRiP [44] on the simulated genomes with parameters described in the S1 Text and CONGA with two values for the C-score (<0.3 and <0.5). We used the above-described list of CNVs as the input candidate set for CONGA and GenomeSTRiP.

To determine true calls, we used >50% reciprocal overlap for the two CNV events (the event in the input event set and the called event) to be considered the same. The overlaps were determined using BEDTools "intersect" [113]. The number of true CNVs were: 1810 deletions

and 1751 duplications for 100 bps–1000 bps; 1680 deletions and 1684 duplications for 1000 bps–10,000 bps; and 1385 deletions and 1532 duplications for 10,000 bps–100,000 bps.

**Down-sampling experiment with real ancient genomes.** We used three relatively high coverage ($\sim$23.3$\times$, $\sim$13.1$\times$ and $\sim$9.6$\times$ respectively) genomes of a Yamnaya culture-related individual from early Bronze Age Karagash (hereafter Yamnaya), Kazakhstan [56], a Saqqaq culture-related individual from Bronze Age Greenland (hereafter Saqqaq) [57], and a 4500-year old East African hunter-gatherer individual from Mota Cave in Ethiopia (hereafter Mota) [58]. Using this data, and the above-described 17,392 deletions and 14,888 duplications of size >1 kbps (see above) as input, we genotyped 2639 deletions and 1972 duplications in Yamnaya (deletion sizes: 1 kbps to 4 Mbps, median = 4 kbps, mean = 23 kbps; duplication sizes: 1 kbps to 28 Mbps, median = 14 kbps, mean = 80 kbps); 1581 deletions and 4097 duplications in Saqqaq (deletion sizes: 1 kbps to 5 Mbps, median = 5 kbps, mean = 17 kbps; duplication sizes: 1 kbps to 28 Mbps, median = 16 kbps, mean = 70 kbps); and 688 deletions and 638 duplications in Mota (deletion sizes: 1 kbps to 130 kbps, median = 4 kbps, mean = 7 kbps; duplication sizes: 1 kbps to 28 Mbps, median = 6 kbps, mean = 82 kbps).

We then randomly down-sampled the BAM files to various depths using Picard Tools [116]: between 16–0.07$\times$ for Yamnaya; 9–0.05$\times$ for Saqqaq; 7–0.03$\times$ for Mota. We note that this down-sampling procedure does not produce the exact targeted depths, which is the reason why we obtain the variable coverages in Fig 4.

For calling deletions we used C-score<0.5. For calling duplications, we called events that fulfilled either of the following conditions (a) C-score<0.5, or (b) C-score<10 and read-pair support >10. Finally, treating the results of the original data as the correct call-set, we calculated TPR (true positive rate) and FDR (false discovery rate) for the down-sampled genomes. We considered CNVs with $\geq$ 50% reciprocal overlap as representing the same event, calculated using BEDTools [113].

**C-score and read-pair cutoffs and minimum CNV size.** We ran CONGA with a range of parameter values for the C-score [0.1–5] and for minimum read-pair support (from 0 support to >30), and using the above-described true event sets as the input candidate set involving medium and large CNVs (1680 deletions and 1684 duplications for 1000 bps–10,000 bps, and 1385 deletions and 1532 duplications for 10,000 bps–100,000 bps).

We used simulation results (Table F in S1 Table) to choose an effective cutoff for calling CNVs. For both deletions and duplications, we decided to use C-score <0.5, which appears to yield a good trade-off between recall and precision. Specifically, in simulations, this cutoff ensured an F-score of >0.5 at 0.1$\times$ for >1 kbps deletions, and superior F-scores at higher coverages (S13 Fig).

In addition, we observed that read-pair support >10 could be useful for identifying duplications in the absence of read-depth support, but only when coverages were $\geq$1$\times$ (Table F in S1 Table and S10 Fig). Moreover, read-pair support was not effective for detecting deletions.

We note that CONGA outputs the C-scores and read-pair counts for all input CNVs. Users can choose alternative cutoffs to increase recall (higher C-scores) or precision (lower C-scores).

The simulation experiments showed that CONGA was not efficient in identifying events <1 kbps. CONGA therefore ignores events <1 kbps under default parameters. This can be modified by the user if needed.

## Analysis of real ancient genomes

**Ancient genome selection and preprocessing.** We selected 71 ancient shotgun-sequenced or whole-genome captured genomes from human skeletal material originating

from West and East Eurasia and from North America (S2 Table). Our sample set belongs to a time range between c.2,800–45,000 years Before Present (BP). Samples from 10 different laboratories were selected in order to study the effects of different data production protocols on deletion genotyping. We also chose genomes with a range of coverage levels (0.04×-26×, median = 3.45×) and that included both UDG-treated and non-UDG-treated libraries. The only capture-produced genome was Bon002 [66], produced using whole-genome hybridization with myBaits (Arbor Biosciences, USA) probes.

Selected ancient genomes were mapped to the human reference genome (hg19, or GRCh37) using BWA aln/samse (0.7.15) [104] with parameters "-n 0.01, -o 2" and disabling the seed with "–l 16500". PCR duplicates were removed using FilterUniqueSAMCons.py [117].

We also removed reads with >10% mismatches to the reference genome, those of size <35 bps, and with <30 mapping quality (MAPQ).

**Candidate CNV call set for real ancient genomes.** Here our goal was to study the properties of deletion variants in ancient genomes and to compare these with SNP variation in terms of demographic history and purifying selection. Polymorphism data sets can suffer from ascertainment bias in downstream evolutionary analyses [88]. A common practice to avoid this bias is to use SNPs ascertained in a population that is an outgroup to the focal populations. We therefore used variants ascertained in modern-day African populations for both calling SNP and deletion variants in our ancient genomes.

In order to create a candidate deletion call set to be used as input to CONGA, we downloaded deletions of size >1000 bps identified among 661 African population (AFR) genomes of the 1000 Genomes Project Phase 3 [52]. When a deletion was located inside the breakpoints of another deletion, we removed the internal one. In addition, for pairs of deletions that had >50% overlap, we filtered out the smaller one. Finally, we filtered out deletion loci with <50% average mappability (see above). This resulted in 11,390 autosomal >1000 bps deletions from 661 AFR genomes.

For downstream analyses, we further filtered these deletions for high mappability ($\geq 0.9$ average mappability) and being derived in the human lineage (see section "Ancestral state determination" below). This left us with 10,002 deletion loci.

**Deletion genotyping in ancient genomes.** We genotyped all the chosen 71 ancient genomes using the 11,390 AFR autosomal deletion data set (>1 kbps with mean 10,735 bps). We used C-score <0.5 as cutoff for calling deletions, and >2 for calling the reference homozygous genotype (0/0). To limit false negatives, C-scores between [0.5–2] were coded as missing (NA). Note that these cutoffs can be modified by the user.

In total, 1,222 deletion loci (12%) out of 10,002 were missing across all the 71 genomes. Of the remaining, 5,467 were genotyped as a deletion in heterozygous or homozygous state in at least one genome. Genotyping rates (non-missing values) in the full dataset was overall 80.0%.

## Analyzing the ancient deletion dataset

We generated a heatmap summarizing deletion copy numbers using the R "gplots" package "heatmap.2" function [118]. Further, we performed principal components analyses (PCA) on the deletion copy number data set (removing missing values) including (a) n = 71, (b) n = 60 (after the first outlier filter), or (c) n = 50 (refined data set) ancient genomes (S4 Fig). PC1 and PC2 values were computed using the R "stats" package "prcomp" function using the default parameters [119]. Second, we investigated deletion frequencies across the same 3 genome sets using multidimensional scaling plots (MDS). These were calculated with parameter "k = 2" with the R "cmdscale" function on a Euclidean distance matrix of deletion frequencies

(without removing NAs). Third, using the same 3 genome sets, we draw hierarchical clustering trees summarizing Manhattan distance matrices, calculated with the R "dist" and "hclust" functions.

These analyses revealed visible outliers in deletion frequency among samples, which collectively we define as the "divergent" genome set (S3(A) Fig; Supplemental Note 3 in S1 Text). We then compared the total number of missing values, average read length, and coverage between the "divergent" genome set (n = 21) and the rest, which we refer to as the "coherent" set (n = 50). Here we used the Mann-Whitney U test with the R "wilcox.test" function, and visualized the data with R utility function "boxplot" [119] (S5A–S5D Fig). We likewise compared average deletion frequencies between UDG-treated and untreated genomes using the Mann-Whitney U test.

## Creating and analyzing the refined deletion data set and the SNP data set

**SNP genotyping in ancient genomes.**   Following the same reasoning as above regarding ascertainment bias, we used the African population as an ascertainment population to create an SNP genotyping set for calling SNPs in the ancient genomes. To create this dataset, we started with all bi-allelic SNPs in the 1000 Genomes Project phase 3 dataset [120] and selected the SNPs with a minor allele frequency greater than zero in 661 African genomes of the 1000 Genomes Project Phase 3. First, all reads in all BAM files were clipped (trimmed) using the trimBam algorithm implemented in BamUtil [121]. Following standard practice [122], we trimmed (a) the end 2 bases of each read for samples prepared with the Uracil-DNA-glycosylase (UDG) protocol, and (b) the end 10 bases of each read for non-UDG samples.

Using these BAM files of the 50 ancient individuals and the above-described SNP list, we generated pseudo-haploid SNP calls at these target SNP positions by randomly selecting one read and recording the allele carried on that read as the genotype. This was performed using the pileupCaller software (https://github.com/stschiff/sequenceTools) on samtools mpileup output (base quality>30 and MAPQ>30) [123].

**Ancestral state determination.**   To polarize deletions and SNP alleles for being ancestral or derived in the human lineage, we mapped loci from hg19 (GRCh37) to panTro6 (chimpanzee) and to panPan2 (bonobo) using the UCSC Genome Browser tool "liftOver" with default parameters [124]. For deletions, we filtered out deletions that did not fully map to either chimpanzee or bonobo reference genomes, as these could represent derived insertions in the human lineage. The remaining deletions could thus be inferred to be alleles that were derived in humans. For SNPs, we removed the positions not represent in either chimpanzee or bonobo reference genomes and assigned the ancestral state as the Pan allele, only if both chimpanzee and bonobo carried same allele. This left us with 32,344,446 SNP positions with derived allele information.

**The refined data set of 50 ancient genomes.**   We removed 21 genomes identified as outliers in both heatmap, PCA and MDS analyses. Next, we genotyped the 8,780 AFR deletions in the remaining 50 genomes. We call this the "refined data set". To investigate its general properties, we plotted the size distribution of deletions, the deletion allele frequency distribution, and the relative frequency distributions of heterozygous over homozygous deletions using the R "graphics" package "hist" function (S14 Fig) [119]. We also plotted relative deletion (homozygous or heterozygous) frequencies of 8,780 deletions for each individual in our refined data set using the R "graphics" package "matplot" function [119].

## Genetic distance and selection analyses using deletions and SNPs

Here our goal was to calculate pairwise genetic distances among the 50 ancient genomes using deletion allele frequencies and using SNPs, and also, to compare the two types of distances.

We calculated distances using the commonly used outgroup-$f_3$ statistics, which measures shared genetic drift between two samples relative to an outgroup, and is implemented as qp3pop in Admixtools v.7.0 [91]. The outgroup-$f_3$ values were calculated for each pair of 50 individuals (a) in the deletion and (b) in the SNP data sets, using the African Yoruba as outgroup in both cases. To convert the deletion data set to eigenstrat format, which Admixtools requires, we encoded the first nucleotide of each deletion as the reference allele, and the alternative allele was randomly assigned among the remaining 3 nucleotides using custom Python script. We thus calculated a pairwise similarity matrix for both data sets. Genetic distances were calculated as 1-$f_3$. Distances were then summarized using multidimensional scaling (MDS) with the "cmdscale" function of R [119] (Fig 5C and 5D and S4 Fig).

We further performed the Mantel test to compare the $f_3$-based similarity matrices calculated using SNPs and deletions. We used the "mantel" function in the R-package "vegan" with parameter "method = spearman" [125].

**Site frequency spectrum (SFS) calculation for deletions and SNPs.** Here our goal was to compare the SFS across deletions and SNPs called in ancient genomes. Because the ancient SNP genotypes are pseudo-haploidized, we performed the same pseudo-haploidization process on the deletion data set. For this, for any heterozygous call in the deletion data set, we randomly assigned either of the homozygous states, using the R "sample" function (i.e., we converted 1's to 0's or 2's with 50% probability). We then counted derived alleles at each locus, for deletions and for SNPs, and divided by the total number of genomes. We removed any loci that were not genotyped in any of the 50 ancient genomes, leaving us with n = 3,472 deletions and n = 57,307 SNPs. We plotted the SFS analysis on both deletions and SNPs using the R "ggplot2" package geom_histogram function [93]. We calculated the Spearman correlation between the deletion size in logarithmic scale and the frequency using the R "stats" package "cor.test" function [119]. We plotted the site frequency spectrum analysis on deletions in high and low coverage genomes using the R "ggplot2" package geom_histogram function [93] (S6 Fig). The threshold is considered to be the median coverage (3.98×).

**Evolutionary conservation.** To measure evolutionary conservation for genes that overlapped deletions, we retrieved non-synonymous (dN) and synonymous (dS) substitution rate estimates between human (GRCh37) and the mouse genome (GRCm38) per gene from Ensembl (v75) via the R package "biomaRt" [126]. We queried 18,112 genes with dN, dS values and calculated the dN/dS (or Ka/Ks) ratio per gene. The ratio for genes with more than one dN or dS values were calculated as the mean dN or dS per gene. We then intersected our deletions with the genes with dN/dS values using BEDTools [113] and found 2,221 Ensembl (v75) human genes. Overall, 34% of the 10,002 derived deletions overlapped with at least one gene. We then collected mouse-human dN/dS ratios (Methods) for these genes (n = 2,221, 0–1.18, median = 0.09, mean = 0.13). For deletions overlapping multiple genes, we calculated the mean dN/dS per deletion. We then divided the deletions in our data set into two groups by the deletion allele frequency: high versus low relative to the median. We plotted the dN/dS ratios of the deletion groups defined above using the R package "ggplot2" and the "geom_boxplot" function [93].

## Comparison with SIFT predictions and temporal change in deletion frequencies

Here our goal was to study deleterious mutation loads per genome in the form of SIFT-predicted harmful SNPs and CONGA-predicted deletions, across the 50 ancient genomes. We used SIFT predictions available in Ensembl (v75) collected via the R package "biomaRt" [126, 127]. We retrieved SIFT predictions of "tolerated" and "deleterious" impact and SIFT scores

for all 1000 Genomes human SNPs from Ensembl, and subsetted the African SNP set used for genotyping the ancient genomes. This resulted in 22,996 SNPs with SIFT predictions. Further, we calculated a ratio representing the total number of SIFT-predicted "deleterious" SNPs over the number of "tolerated" SNPs, for each of the 50 individuals. In addition, we calculated the total CONGA-predicted deletion length and the total number of genes overlapping CONGA-predicted deletions per individual, ignoring homozygous or heterozygous state. We plotted these three mutation load scores, i.e. SIFT-predicted deleterious/tolerated ratios per individual, the number of affected genes, and the total deletion length, using R base function "plot" (S7 Fig) [119]. We further estimated pairwise correlations between the three scores, fitting the values into a linear model using the R "lm" function and calculating Spearman's rank correlation. We plotted the linear models using the R base function "pairs" (B in S8 Fig) [119].

We finally tested whether the mean deletion allele frequency changed over time by fitting the values in a linear model using the R "lm" function (S9 Fig).

## Supporting information

**S1 Text. Command lines and supplementary notes (1-4).**
(PDF)

**S1 Fig. Length distribution of CNVs inserted into the simulated genomes.** The total number of CNVs inserted into a genome ("counts") is shown at the top of each graph. We used Varsim to insert these CNVs into each genome, yielding three genomes in total (for short, medium and large CNVs).
(TIF)

**S2 Fig. Precision—Recall plots for simulations.** Precision-Recall curves for deletion (A) and duplication (B) predictions of CONGA, GenomeSTRiP, FREEC, and CNVnator using coverages of 0.05×, 0.1×, 0.5×, 1× and 5×. mrCaNaVaR was used only in the analysis of large variants.
(TIF)

**S3 Fig. Heatmaps of CONGA-genotyped n = 10,002 human-derived deletions across ancient genomes.** The color key includes 0 (gray) for reference allele, 1 (green) for heterozygous, 2 (magenta) for homozygous state and NA (white) for missing value. (A) Heatmap of deletions per genome on the raw dataset (n = 71 genomes). (B) Heatmap of deletions per genome on the refined dataset (with n = 50 genomes after removing divergent genomes).
(TIF)

**S4 Fig. Multivariate analysis of deletion frequencies reveal outlier genomes.** Left panels: Multidimensional scaling plots (MDS) calculated with k = 2 using the R "cmdscale" function on a Euclidean distance matrix of deletion frequencies. Middle panels: Principal component analysis plots (PCA) summarizing deletion frequencies after removing any NAs. Right panels: Hierarchical clustering trees summarizing Manhattan distance matrices, calculated using the R "dist" and "hclust" functions. The color codes indicate the laboratory-of-origin of each genome, shown in the legend of the top right panel. (A) Results based on the full dataset with 10,002 human-derived deletions (n = 8,780 genotyped in any state in at least one genome) and n = 71 genomes. In the PCA we use nD = 580 deletions after removing loci with at least one missing value. (B) Results based on n = 60 genomes after removing 11 outlier genomes (and nD = 3,460 deletions in the PCA). (C) Results based on n = 50 genomes after removing 21

outlier genomes (and nD = 3,472 deletions in the PCA). We note that the MDS here differs from that shown in Fig 5, in that the latter is calculated using outgroup-f3 statistics.
(TIF)

**S5 Fig. Technical comparisons of divergent (n = 21) and coherent (n = 50) genome sets.** This is defined based on their deletion profiles (S3 and S4 Figs) (A) Boxplots of the average read length per genome (Wilcoxon-rank sum test, P<0.001). (B) Boxplots of mean depths of coverage (Wilcoxon rank sum test, P = 0.014). (C) Boxplots of the number of missing values per genome (Wilcoxon-rank sum test, P<0.001). (D) Boxplots of the mean deletion allele frequencies per UDG-treated and not UDG-treated genomes. We observe no significant difference between the distributions (Wilcoxon-rank sum test, P = 0.58).
(TIF)

**S6 Fig. Site-frequency spectra of deletions genotyped in low and high coverage genomes.** Left panel represents the SFS of n = 25 below-median coverage genomes and right panel shows the SFS of n = 25 above-median coverage genomes. The median coverage value was 3.98×. We found no significant difference between the two SFS distributions (Kolmogorov-Smirnov test $\rho$ = 0.27).
(TIF)

**S7 Fig. Deleterious load estimates among 50 ancient genomes.** In all three panels, the x-axis represents a deleterious load-related statistic and the y-axis shows the ancient individuals. (A) Deleterious load based on SIFT-estimated SNP effects per individual. The x-axis represents the number of "deleterious" SNPs over the number of "tolerated" SNPs. (B) CONGA-estimated total deletion length in kb per individual, using the Final CNV call-set. (C) The number of genes that overlap with CONGA-estimated deletions. In panels B and C, heterozygous and homozygous calls were counted once. In panel C, the most affected individuals in terms of the number of gene overlaps are RISE497 (Russia, 2nd millennium BCE), DA380 (Turkmenistan, 4th millennium BCE), RISE675 (Russia, 3rd millennium BCE), and Chan (Iberia, 8th millennium BCE). We observed that these individuals were around 50% more affected than the rest.
(TIF)

**S8 Fig. Correlations between SNP- and deletion-based deleterious load estimates in 50 ancient genomes.** (A) From left to right: histograms of the number of SIFT-predicted "deleterious" SNPs over "tolerated" SNPs per genome, CONGA-predicted total deletion length in kb per genome, and the number of genes that overlap with CONGA-predicted deletions per genome. (B) Correlations between each variable. The RHS triangle shows the scatter plots between two variables, and the LHS triangle shows the Spearman rank correlation estimates. The significance of the $\rho$'s are also shown. **** $\rho$<0.0001, ns: non-significant.
(TIF)

**S9 Fig. Scatter plot of the mean allele frequencies per genome (n = 50) vs. age of the genomes (calibrated years before present).** Red line represents the linear model. We found that there is no significant correlation between the age of the individuals and the mean allele frequency (Spearman's rank correlation $\rho$ = -0.12, P = 0.41).
(TIF)

**S10 Fig. Effect of minimum read-pair support on the F-Score for duplications using various depths of coverage values in simulated genomes.** Medium sized CNVs are between 1,000 bps to 10,000 bps and large CNVs are between 10,000 bps and 100,000 bps. Here, we used a relaxed C-score threshold of 10 in order to observe the effect of read-pair support only.

The figure shows that read-pair support is effective when the coverage is above 0.5x and also when the duplication sizes are larger.
(TIF)

**S11 Fig. IGV visualization of two high scoring (i.e., high likelihood) deletions and duplications predicted by CONGA.** The events displayed in the upper panels were detected in a modern-day human genome (NA07051: an ∼8 kbp deletion within chr7:16,169,440-16,177,556 and a ∼4 kbp duplication within chr7:22,496-26,553) and those in the lower panels in an ancient genome (RISE98: an ∼17 kbp deletion within chr6:32,506,809-32,524,264 and a ∼6 kbp duplication within chr1:1,520,604-1,526,959). The candidate CNV list used for genotyping was the long read CNV dataset described in Methods. Deducing the CNVs is straightforward with the modern-day genome data, however, it is less straightforward to distinguish these variations in ancient read data, especially for duplications. Note that this is one of the sample scenarios and we emphasize that a large number of CNVs identified in ancient genomes suffer from the same issue.
(TIF)

**S12 Fig. A sample deletion missed due to poor signal.** An inserted deletion in a simulated ancient genome at 1× depth of coverage. The event breakpoint is chr22:39,386,521-39,391,930. CONGA missed this deletion due to the poor signal.
(TIF)

**S13 Fig. The effect of C-score on F-Scores.** The figure shows the effect of C-score on F-Scores of deletions (A) and duplications (B) for 0.1× and 5× depths of coverages in simulated genomes with medium sized CNVs embedded. Here, we did not use read-pair support or mappability filtering, in order to only test the effect of the C-score threshold. The C-score is calculated using read-depth information.
(TIF)

**S14 Fig. General characteristics of deletions in the refined dataset (n = 50 genomes and n = 8,780 deletions, obtained after applying ancestry state filters and removing outlier genomes).** (A) Size distribution of the deletions in logarithmic scale. (B) The distribution of the deletion allele frequency (i.e. the proportion of deletion alleles across the 8,780 loci per genome) among the 50 genomes. (C) The distribution of the relative frequency of observed heterozygous (0/1) deletions over homozygous (1/1) deletions observed in our dataset. (D) The plot of relative deletion frequencies called heterozygous (red lines) and homozygous (blue lines) among 8,780 deletions, for each of the n = 50 ancient genomes in the refined dataset (after applying additional ancestry state filters and removing outlier genomes).
(TIF)

**S15 Fig. Split-read approach to emulate paired-end using single-end reads.** We use short-read Illumina mappings in a BAM file as input. We split each discordant read (whose mapping quality is larger than the given threshold and does not overlap with a known satellite) from the middle, keeping the initial mapping as one element and the other subsequence (split segment) as the second element of a pair. We remap the split segment to the reference genome, and evaluate the position and the orientation of both reads to identify the presence of putative CNVs.
(TIF)

**S1 Table. Performance comparison of multiple tools and CONGA performance analysis.** Table A shows the CNV predictions of CONGA, GenomeSTRiP, FREEC, CNVnator and mrCaNaVaR on simulated genomes at depths 0.05×, 0.1×, 0.5×, 1× and 5× for deletions and

duplications of multiple CNV size intervals including 100 bps–1 kbps (small), 1 kbps–10 kbps (medium) and 10 kbps–100 kbps (large). Here, "T" and "F" refer to correct and incorrect predictions respectively, "Miss" is the number of missed true events, "Recall" (TPR) is the true positive rate, and "FDR" is false discovery rate (1—"Precision") for each run. The F-Score is calculated as (2 * Precision * Recall) / (Precision + Recall). Note that for CONGA, we included the performance for both C-score<0.5 and C-score<0.3. The table also contains a comparison between CONGA and GenomeSTRiP predictions.

Table B shows a comparison between CONGA and GenomeSTRiP predictions on simulated genomes at depths 0.05×, 0.1×, 0.5×, 1× and 5× for deletions and duplications of 1 kbps–10 kbps (medium) and 10 kbps–100 kbps (large) CNV size intervals.

Table C shows the copy-number (homozygous or heterozygous) predictions of CONGA on simulated genomes at depths 0.05×, 0.1×, 0.5×, 1× and 5× for deletions and duplications of multiple CNV size intervals including 100 bps–1 kbps (small), 1 kbps–10 kbps (medium) and 10 kbps–100 kbps (large). Here, "T" and "F" refer to correct and incorrect predictions respectively, "Miss" is the number of missed true events, "Recall" (TPR) is the true positive rate, and "FDR" is false discovery rate (1—"Precision") for each run. The F-Score is calculated as (2 * Precision * Recall) / (Precision + Recall). Note that for CONGA, we included the performance for both C-score<0.5 and C-score<0.3.

Table D shows deletion and duplication predictions of CONGA using Mota, Saqqaq and Yamnaya genomes down-sampled to various depths from their original coverages of 9.6×, 13.1× and 23.3×, respectively. Here, "T" and "F" refer to correct and incorrect predictions respectively, "Miss" is the number of missed true events, "Recall" (TPR) is the true positive rate, and "FDR" is false discovery rate (1—"Precision") for each run. The F-Score is calculated as (2 * Precision * Recall) / (Precision + Recall). We calculated "True", "False", "Miss", "Recall", "Precision", FDR and F-Score of down-sampled genomes assuming that our CONGA-based predictions with the original genomes (full data) reflect the ground truth. These predictions, in turn, were made using modern-day CNVs as candidate CNV list. The purpose of the experiment was to evaluate accuracy at lower coverage relative to the full data.

Table E shows CONGA's running time and memory consumption on genomes of various depths of coverage calculated using the down-sampled 23× Yamnaya genome (with coverages between 23× and 0.07×) as well as a comparison of CONGA, GenomeSTRiP, FREEC and CNVnator using a 5× simulated genome.

Table F shows the results of CONGA runs on simulated genomes at a range of parameters, performed in order to determine the optimum parameters to be used. We tested multiple parameter combinations of C-Score, minimum read-pair support, mappability and minimum mapping quality (MAPQ) using simulated genomes.

(XLSX)

**S2 Table. Information on the ancient genomes used in this study.** "ID_publication" refers to the ID of the genome used in the original publication, and "Sample_ID" refers to the ID used in this study. "Average Date (BP)" is the average date in years before present. "Lab PI" refers to the senior author of the study. "Included in final analysis" shows the genomes included in the list of 50 after removing divergent (outlier) genomes.

(XLSX)

## Acknowledgments

The authors would like to thank Gözde Zeliha Turan for her suggestions with mappability data and Kıvılcım Başak Vural for technical support.

## Author Contributions

**Conceptualization:** Arda Söylev, Can Alkan, Mehmet Somel.

**Data curation:** Arda Söylev, Sevim Seda Çokoglu, Dilek Koptekin, Mehmet Somel.

**Formal analysis:** Arda Söylev, Sevim Seda Çokoglu, Dilek Koptekin.

**Funding acquisition:** Mehmet Somel.

**Investigation:** Arda Söylev, Sevim Seda Çokoglu, Dilek Koptekin, Mehmet Somel.

**Methodology:** Arda Söylev, Sevim Seda Çokoglu, Mehmet Somel.

**Project administration:** Mehmet Somel.

**Resources:** Dilek Koptekin, Mehmet Somel.

**Software:** Arda Söylev, Can Alkan.

**Supervision:** Can Alkan, Mehmet Somel.

**Validation:** Arda Söylev, Sevim Seda Çokoglu, Dilek Koptekin.

**Visualization:** Arda Söylev, Sevim Seda Çokoglu, Dilek Koptekin, Mehmet Somel.

**Writing – original draft:** Arda Söylev, Sevim Seda Çokoglu, Dilek Koptekin, Can Alkan, Mehmet Somel.

**Writing – review & editing:** Arda Söylev, Sevim Seda Çokoglu, Dilek Koptekin, Mehmet Somel.

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
