## [Decision Letter · Decision Letter 0]

6 Oct 2022

Dear Dr. Soylev,

Thank you very much for submitting your manuscript "CONGA: Copy number variation genotyping in ancient genomes and low-coverage sequencing data" for consideration at PLOS Computational Biology.

As with all papers reviewed by the journal, your manuscript was reviewed by members of the editorial board and by several independent reviewers. In light of the reviews (below this email), we would like to invite the resubmission of a significantly-revised version that takes into account the reviewers' comments.

As you will see, all three reviewers appreciated your manuscript and the novelty and importance of your approach. In addition to the specific remarks that reviewers made, which I believe will improve your manuscript, two of the reviewers note that they were not able to compile and test CONGA. This is very important for a computational biology paper, and I ask you to please insure that the software can be compiled and used before resubmission, and address this question very clearly in the revision.

We cannot make any decision about publication until we have seen the revised manuscript and your response to the reviewers' comments. Your revised manuscript is also likely to be sent to reviewers for further evaluation.

Sincerely,

Marc Robinson-Rechavi

Academic Editor

PLOS Computational Biology

Sushmita Roy

Section Editor

PLOS Computational Biology

As you will see, all three reviewers appreciated your manuscript and the novelty and importance of your approach. In addition to the specific remarks that reviewers made, which I believe will improve your manuscript, two of the reviewers note that they were not able to compile and test CONGA. This is very important for a computational biology paper, and I ask you to please insure that the software can be compiled and used before resubmission, and address this question very clearly in the revision.

Reviewer's Responses to Questions

**Comments to the Authors:**

Reviewer #1: uploaded as an attachment

Reviewer #2: CONGA

Summary:

The authors propose CONGA, a specialized tool for the genotyping of copy-number variation in ancient genomes, which come with a plethora of challenges owing to low coverage, sample damage, and issues in library preparation. CONGA uses any given set of CNV calls as a truth set for genotyping using read-depth and split-read methodologies.

The authors evaluate the performance of their algorithm on simulated data, down-sampled real genomes, as well 71 ancient genomes, in a manner spanning a wide range of coverages and diverse ancestry.

The paper is written well - its question is clear, the goals of each section are appropriate, described effectively and concisely.

The authors do an excellent job of describing not only the strengths and applications of their algorithm, but also its faults, caveats, and areas of underperformance. The paper highlights the usability of CONGA while explaining clearly areas the tool is not suitable for analysis (such as genotyping ancient genomes in <5x coverage).

Additionally, the algorithm performs as expected. True-positive rate, as well as false-discovery rate largely scale with coverage. The algorithm tends to perform better for deletions than other types of CNVs, and CONGA is most performant for variants > 1kb as expected of short-read datasets.

The results are clear and the tool looks useful for those who are interested in genotyping CNVs in low-coverage ancient genomes. Overall, CONGA is deserving of publication in this journal without revision.

Major:

None

Minor:

Page 2, Line 83 - The CONGA algorithm is initially described in terms of read-depth and split-reads. However, a small line should be inserted here to quickly inform of the reader of how CONGA uses this information differently from already available read-depth and split-read callers.

Page 3, Line 119 - The authors title this section 'copy number predictions of CNVs' but note that CONGA does not evaluate >= 3 copies. This section should be renamed, as many working on CNVs may assume more capability here as opposed to simply detecting homozygous vs homozygous variation.

The authors mention that their input callset was determined across 4 datasets by choosing non-overlapping variants, but some polymorphic CNVs in the population may have overlapping breakpoints and thus such a heuristic may filter out genotype-able CNVs. It would be worth attempting to decide on a way to keep some of these CNVs in the input callset, perhaps based on some metric of reciprocal overlap.

Reviewer #3: In attached file.

**Have the authors made all data and (if applicable) computational code underlying the findings in their manuscript fully available?**

Reviewer #1: Yes

Reviewer #2: Yes

Reviewer #3: Yes

PLOS authors have the option to publish the peer review history of their article (what does this mean?). If published, this will include your full peer review and any attached files.

Reviewer #1: No

Reviewer #2: No

Reviewer #3: No
---

## [Decision Letter · Decision Letter 1]

18 Nov 2022

Dear Dr. Soylev,

Thank you very much for submitting your manuscript "CONGA: Copy number variation genotyping in ancient genomes and low-coverage sequencing data" for consideration at PLOS Computational Biology. As with all papers reviewed by the journal, your manuscript was reviewed by members of the editorial board and by several independent reviewers. The reviewers appreciated the attention to an important topic. Based on the reviews, we are likely to accept this manuscript for publication, providing that you modify the manuscript according to the review recommendations.

Like the reviewer, I have not been able to compile your software. I checked with the reviewer, and they tried on Linux, as recommended in the instructions. While it is acceptable that the software compiles on Linux and not on other OSes, I cannot accept it for publication in PLOS Comp Biol if neither reviewers nor myself have been able to compile and run the code in at least one environment.

Sincerely,

Marc Robinson-Rechavi

Academic Editor

PLOS Computational Biology

Sushmita Roy

Section Editor

PLOS Computational Biology

Like the reviewer, I have not been able to compile your software. I checked with the reviewer, and they tried on Linux, as recommended in the instructions. While it is acceptable that the software compiles on Linux and not on other OSes, I cannot accept it for publication in PLOS Comp Biol if neither reviewers nor myself have been able to compile and run the code in at least one environment.

Reviewer's Responses to Questions

**Comments to the Authors:**

Reviewer #1: The authors did a great job revising the manuscript incorporating the addressed points. The manuscript has gained in quality.

Unfortunately, the compilation of the code is still challenging, and I did not succeed with the given information. I am not sure if interested researchers will make this effort. Having a conda installation or a universal binary would be helpful. In the following I try to list the problems I encountered to compile the code:

System without root/sudo access:

Did not figure out how to get libbz2 and liblzma installed without root access.

System with root/sudo access (libcurl is missing):

$ sudo apt-get install libbz2-dev

$ sudo apt-get install liblzma-dev

$ make -C htslib/

…

gcc -g -Wall -O2 -I. -c -o hfile_libcurl.o hfile_libcurl.c

hfile_libcurl.c:46:10: fatal error: curl/curl.h: No such file or directory

46 | #include <curl curl.h="">

| ^~~~~~~~~~~~~

compilation terminated.

make: *** [Makefile:132: hfile_libcurl.o] Error 1

System with root/sudo access and cheating (disabling libcurl):

$ sudo apt-get install libbz2-dev

$ sudo apt-get install liblzma-dev

$ wget https://github.com/samtools/htslib/releases/download/1.16/htslib-1.16.tar.bz2

$ tar --bzip2 -xvf htslib-1.16.tar.bz2

$ cd htslib-1.16

$ ./configure ## libcurl is disabled

$ make

$ cd ../

$ git clone https://github.com/asylvz/CONGA --recursive

$ cd CONGA

$ mv htslib htslib2

$ ln -s ../htslib-1.16 htslib

$ make -C sonic

$ sed -i ‘s/-lcurl //g’ Makefile ## remove libcurl dependency

$ make</curl>

**Have the authors made all data and (if applicable) computational code underlying the findings in their manuscript fully available?**

Reviewer #1: Yes

PLOS authors have the option to publish the peer review history of their article (what does this mean?). If published, this will include your full peer review and any attached files.

Reviewer #1: No

Figure Files:

Data Requirements:

Reproducibility:

References:

---

## [Editor Report · Decision Letter 2]

3 Dec 2022

Dear Dr. Soylev,

We are pleased to inform you that your manuscript 'CONGA: Copy number variation genotyping in ancient genomes and low-coverage sequencing data' has been provisionally accepted for publication in PLOS Computational Biology.

Best regards,

Marc Robinson-Rechavi

Academic Editor

PLOS Computational Biology

Sushmita Roy

Section Editor

PLOS Computational Biology

---

## [Editor Report · Acceptance letter]

9 Dec 2022

PCOMPBIOL-D-22-01205R2 

CONGA: Copy number variation genotyping in ancient genomes and low-coverage sequencing data

Dear Dr Söylev,

I am pleased to inform you that your manuscript has been formally accepted for publication in PLOS Computational Biology. Your manuscript is now with our production department and you will be notified of the publication date in due course.

With kind regards,

Anita Estes
